# Conserved Expression and Functionality of Furin between Chickens and Ducks as an Activating Protease of Highly Pathogenic Avian Influenza Virus Hemagglutinins

Anja C. M. de Bruin,[a] Monique I. Spronken,[a] Theo M. Bestebroer,[a] Ron A. M. Fouchier,[a] Mathilde Richard[a]

[a]Department of Viroscience, Erasmus Medical Center, Rotterdam, the Netherlands

**ABSTRACT** Highly pathogenic avian influenza viruses (HPAIVs) typically emerge from low-pathogenic avian influenza viruses (LPAIVs) of the H5 and H7 subtypes upon spillover from wild aquatic birds into poultry. The conversion from LPAIV to HPAIV is characterized by the acquisition of a multibasic cleavage site (MBCS) at the proteolytic cleavage site in the viral binding and fusion protein, hemagglutinin (HA), resulting in cleavage and activation of HA by ubiquitously expressed furin-like proteases. The ensuing HPAIVs disseminate systemically in gallinaceous poultry, are endotheliotropic, and cause hemorrhagic disease with high mortality. HPAIV infections in wild aquatic birds are generally milder, often asymptomatic, and generally not associated with systemic dissemination nor endotheliotropic. As MBCS cleavage by host proteases is the main virulence determinant of HPAIVs in poultry, we set out to determine whether cleavage of HPAIV HA by host proteases might influence the observed species-specific pathogenesis and tropism. Here, we sequenced, cloned, and characterized the expression and functionality of duck furin. The furin sequence was strongly conserved between chickens and ducks, and duck furin cleaved HPAIV and tetrabasic HA in an overexpression system, confirming its functionality. Furin was expressed ubiquitously and to similar extents in duck and chicken tissues, including in primary duck endothelial cells, which sustained multicycle replication of H5N1 HPAIV but not LPAIVs. In conclusion, differences in furin-like protease biology between wild aquatic birds and gallinaceous poultry are unlikely to largely determine the stark differences observed in species-specific pathogenesis of HPAIVs.

**IMPORTANCE** HPAIV outbreaks are a global concern due to the health risks for poultry, wildlife, and humans and their major economic impact. The number of LPAIV-to-HPAIV conversions, which is associated with spillover from wild birds to poultry, has been increasing over recent decades. Furthermore, H5 HPAIVs from the A/goose/Guangdong/1/96 lineage have been circulating in migratory birds, causing increasingly frequent epizootics in poultry and wild birds. Milder symptoms in migratory birds allow for dispersion of HPAIVs over long distances, justifying the importance of understanding the pathogenesis of HPAIVs in wild birds. Here, we examined whether host proteases are a likely candidate to explain some differences in the degree of HPAIV systemic dissemination between avian species. This is the first report to show that furin function and expression is comparable between chickens and ducks, which renders the hypothesis unlikely that furin-like protease differences influence the HPAIV species-specific pathogenesis and tropism.

**KEYWORDS** highly pathogenic avian influenza virus, furin, PCSK3, PC5/6, proteases, hemagglutinin, multibasic cleavage site, duck, chicken

Low-pathogenic avian influenza viruses (LPAIVs) circulate in wild aquatic birds (*Anseriformes* [e.g., ducks and geese] and *Charadriiformes* [e.g., gulls]) (1), in which they generally cause asymptomatic infections (2). LPAIVs primarily replicate

Address correspondence to Mathilde Richard, m.richard@erasmusmc.nl.

The authors declare no conflict of interest.

in the gastrointestinal tract, mainly resulting in fecal shedding and presumed fecal-oral transmission. Influenza A viruses are classified based on the antigenic properties of the two surface glycoproteins, the hemagglutinin (HA), mediating receptor binding and membrane fusion, and the neuraminidase (NA), mediating virion release. To date, 16 hemagglutinin (H1 to H16) and nine neuraminidase (N1 to N9) antigenically distinct subtypes have been detected within the wild bird reservoir (3). Upon spillover to terrestrial poultry, H5 and H7 LPAIVs can convert to highly pathogenic avian influenza viruses (HPAIVs) (4–11). HPAIVs cause systemic endotheliotropic hemorrhagic disease in gallinaceous poultry, with mortality rates reaching 100% (12). In wild aquatic birds, infections with HPAIVs are generally milder, not hemorrhagic, nor endotheliotropic (12–15). However, HPAIVs from the H5 A/goose/Guangdong/1/96 lineage (Gs/Gd) can cause systemic disease in wild birds which can be fatal in part due to severe neurological complications (13, 16). Their disease severity depends on the bird species, viral strain, age, and infection history (12, 13, 17–21). Still, the disease manifestations presented by Gs/Gd HPAIV-infected wild aquatic birds are distinct from those in poultry, as endotheliotropism and hemorrhage are seldom observed (14, 16, 22, 23). The reasons behind this stark difference in HPAIV pathogenesis between gallinaceous poultry and wild aquatic birds have not been fully elucidated, but differences in the induction of antiviral and inflammatory responses are thought to be important (24–28).

An essential step in the AIV replication cycle is the posttranslational proteolytic cleavage of the HA precursor HA0 into the disulfide-bound HA1 and HA2 subunits (29). Cleavage enables the low-pH-induced conformational change of HA and liberates the fusion peptide, located at the N-terminus of HA2, so that it is accessible to mediate fusion of the viral and endosomal membranes, which is necessary for release of the genome into the cytoplasm (30, 31). LPAIVs contain a monobasic cleavage site, with one arginine (R) or one lysine (K), whereas HPAIVs contain a multibasic cleavage site (MBCS) with consensus motif R-X-R/K-R (12, 32, 33). The MBCS is the main virulence determinant of AIVs in gallinaceous poultry. The HA of LPAIVs is activated by tissue-specific trypsin-like proteases, restricting virus replication to the respiratory and intestinal tracts (12, 34). In contrast, HPAIV HAs are cleaved by ubiquitous furin-like proteases, which allow for virus replication in a plethora of different tissues and consequently systemic virus dissemination in gallinaceous poultry (35). In contrast, the presence of an MBCS does not necessarily lead to widespread replication of AIVs in wild birds (36), except for some HPAIVs from the Gs/Gd lineage (13, 16, 22). Therefore, the difference in HPAIV tropism between poultry and wild aquatic birds might also be related to a certain extent to expression and activity differences of furin-like proteases.

Furin-like proteases are proprotein convertases subtilisin-kexin (PC/PCSK), of which nine have been identified, i.e., PCSK1 to -9 (reviewed in reference 37). The PC family comprises transmembrane and secreted proteases that occupy specific subcellular residencies along the constitutive secretory pathway. PCs contain an N-terminal signal peptide for endoplasmic reticulum targeting and an inhibitory prodomain that governs correct protein folding and enzyme activation. The secretory pathway becomes progressively more acidic, which determines PC activation, as PCs are activated by autoproteolytic cleavage of the prodomain in a pH-dependent manner (38, 39). The prodomain is succeeded by a highly conserved catalytic domain harboring the catalytic triad consisting of the amino acids D, H, S, and an oxyanion-hole-forming N. This domain is followed by a cysteine-rich domain and an optional transmembrane domain. The cytoplasmic tail contains intrinsic sorting signals that govern localization to specific organelles. The recognition sequence of PCSK1 to -7 is R-X-R/K-R (37). Human furin (PCSK3) has been shown to efficiently cleave HPAIV HA, whereas PC1/3 (PCSK1), PC2 (PCSK2), PC4 (PCSK4), PC5/6 (PCSK5), PACE4 (PCSK6), and PC7 (PCSK7) do so to a lesser extent or not at all, depending on the applied method (40–43). Despite birds being the original hosts of HPAIVs, most studies on the role of furin-like proteases in HA cleavage have been performed in mammalian systems (40–44). Therefore, limited data are available on avian PCs. Genes encoding furin, PC5/6, PC7, and PACE4 have been identified in chickens (*Gallus domesticus*),

and chicken furin has been cloned and expressed exogenously to confirm cleavage of HPAIV H7 HA (45) and mutant versions of H9 HA (46). Furthermore, it has been shown that duck hepatitis B virus requires cleavage by a furin-like protease present in duck hepatocytes (47). Nevertheless, information on furin-like proteases in the context of HPAIV HA activation in wild aquatic birds and whether differences in PC expression and functionality might influence the species-specific pathogenesis of HPAIVs is missing.

Here, we have sequenced and cloned the open reading frame (ORF) of furin from domestic ducks (*Anas platyrhynchos domesticus*) in order to confirm that duck furin can cleave HPAIV HA upon exogenous expression. Furthermore, furin mRNA expression levels were comparable between chickens and ducks in an array of different adult tissues. Finally, as systemic dissemination of HPAIVs in chickens is facilitated by virus replication in the endothelium, we confirmed mRNA PC expression and HPAIV replication in primary duck endothelial cells. In conclusion, we showed that the expression and functionality of furin is conserved between the model species for gallinaceous poultry, chickens, and wild aquatic birds, domestic ducks, and is therefore not likely to contribute largely to the differences in HPAIV tropism and pathogenesis between avian hosts.

## RESULTS

**Duck and chicken furin open reading frames have high sequence identity.** We set out to sequence the furin ORF of the domestic duck, as only partial or predicted sequences were available prior to the onset of this research (e.g., XM_021272973). The ORF sequence was determined by Sanger sequencing from the mRNA of late-stage embryonic lung and/or bone marrow tissues from pooled male and female donors, leading to the detection of synonymous polymorphisms (data not shown). The resulting 789-amino-acid-long furin protein had high identity with its chicken (96%) and to a lesser extent human (79%) counterparts (Fig. 1). Embryonic mallard tissues were sequenced as well to exclude an alteration of the furin ORF following domestication, which yielded an identical amino acid sequence to that of domestic duck furin. Based on sequence identity with human furin, we observed that the duck furin ORF contained the conserved catalytic domain harboring the catalytic triad. Based on the localization of specific features in human furin, we could also identify the inhibitory prodomain with two internal cleavage sites, R71 and R108 (amino acid numbering as in the duck ORF), transmembrane domain, and a variable cytoplasmic domain containing sorting signals (Fig. 1). As can be inferred from the amino acid identity percentages, major differences existed between avian and human furin ORFs, whereas the differences between duck and chicken furin ORF are small. The cysteine topology in the cysteine-rich domain (Fig. 1, amino acids 590 to 678) remained conserved between the species, whereas the largest variation was present in the region preceding the transmembrane domain. The cytoplasmic domain was mostly conserved between human and avian furin, except for a slight variation in the acidic cluster that governs localization of furin to the trans-Golgi network (TGN). However, the total number of acidic residues D and E remained the same. The sorting signal required for basolateral sorting of furin following exit from the TGN in polarized cells contained one acidic residue less in avian compared to human furin.

**Duck furin cleaves MBCS of HPAIVs and tetrabasic MBCS motifs.** In order to assess the functionality of duck furin in the context of HA cleavage, the duck furin ORF was cloned into a pCAGGs expression vector and coexpressed with various HA proteins in the furin-deficient LoVo cell line (50) (Fig. 2A). The experiments were performed using the HA from the A/Indonesia/5/05 H5N1 HPAIV, and as comparison human and chicken furin were included. First, equal levels of furin expression upon transfection of furin from different species were confirmed by immunoblotting (Fig. 2B). A shift in protein size was observed between human and avian furin, which might be attributed to the different amino acid composition. HA cleavage was assessed via western blotting using a polyclonal serum that primarily targeted HA1. The HA proprotein HA0 was detected at a height corresponding to a molecular weight of 75 kDa, and cleaved HA1 was detected at 50 kDa. Background cleavage in LoVo cells was detected for the negative-control HA-ΔMBCS (HPAIV HA from which the MBCS was reverted to the H5 LPAIV consensus RETR),

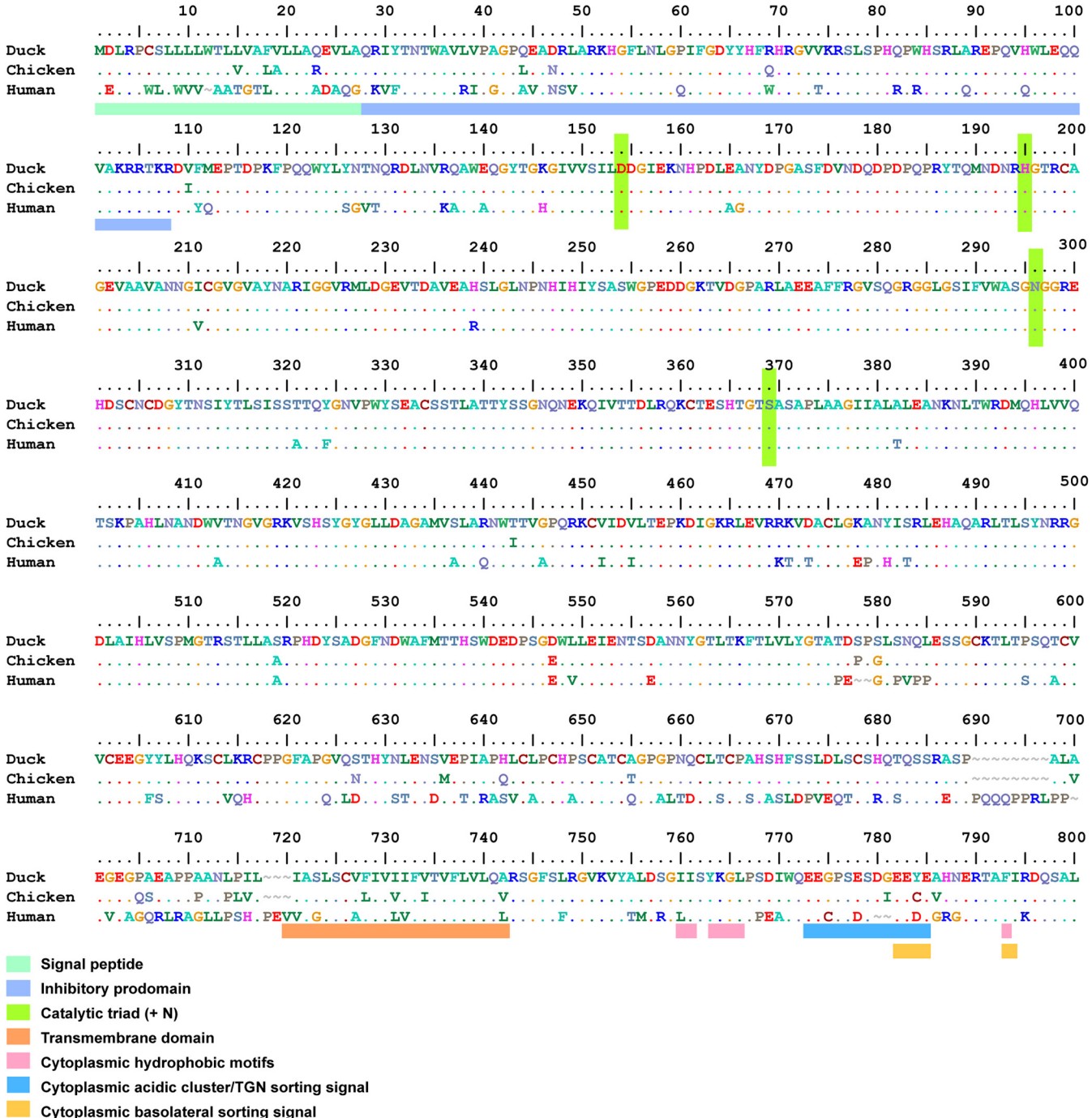

**FIG 1** Comparison of duck, chicken, and human furin ORFs, based on their alignments of the newly obtained sequences of domestic duck, human (NM_002569.4), and chicken (NM_204715.2) furin ORFs. The legend indicates peptide domain features, based on the human furin annotation (37, 48, 49). TGN; trans-Golgi network.

and an overall decrease in HA protein expression was observed when HA was cotransfected with avian furins (Fig. 2C). Treatment of cell pellets with *N*-tosyl-L-phenylalanine chloromethyl ketone (TPCK)-treated trypsin resulted in a reduction of the HA0 and increase in HA1 band intensities, indicating HA cleavage (Fig. 2C and E). As expected, no additional cleavage of the HA-ΔMBCS was detected upon transcomplementation with human, chicken, or duck furin. In contrast, the wild-type HPAIV HA was fully cleaved following transcomplementation with furin from all three species (Fig. 2C and E), demonstrating that HPAIV HA can be cleaved by duck furin in an overexpression system.

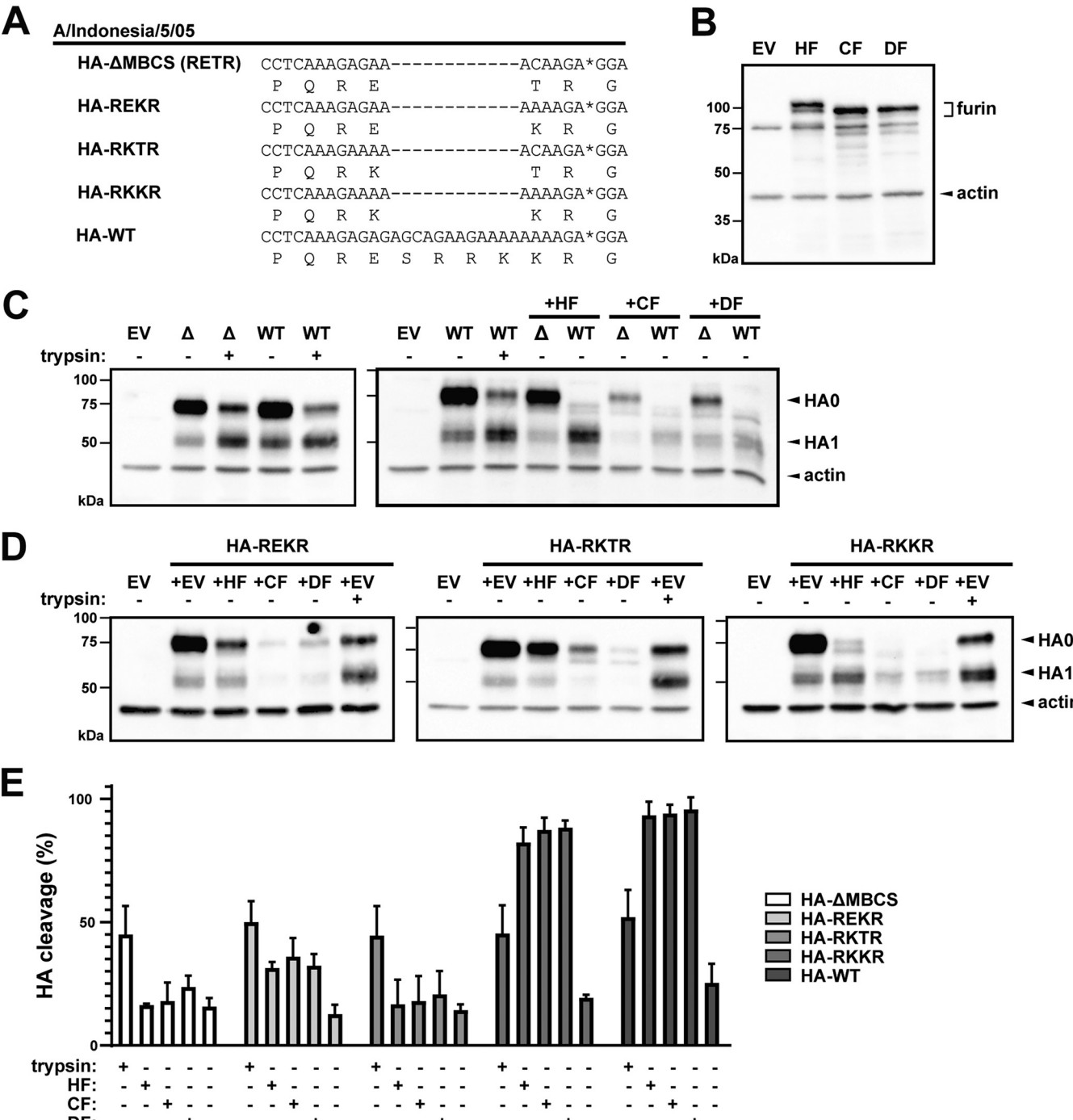

**FIG 2** HAs with tetrabasic and wild-type HPAIV MBCS are cleaved by duck furin. (A) Alignment of cleavage site mutants in the H5 HA from HPAIV A/Indonesia/5/05. HA1 and HA2 are separated by an asterisk. (B) Furin-deficient LoVo cells were transfected with empty vector (EV) or plasmids coding for human (HF), chicken (CF), or duck furin (DF). Cells were harvested and lysed for western blot analysis targeting furin at 24 h posttransfection. Proteins were deglycosylated and size-separated on a 10% reducing and denaturing SDS-PAGE gel. Furin was detected using a rabbit monoclonal antibody. Beta-actin was included as loading control. Protein size in kilodaltons was assessed by running the samples alongside a ladder. Representative blots of three independent experiments are depicted. (C) LoVo cells were cotransfected with plasmids coding for HF, CF, or DF and either HA-WT (WT) or HA-ΔMBCS (Δ). Cells were treated with TPCK-trypsin (2.5 μg/mL) for 10 min as positive control and EV was transfected as negative control. Cells were harvested and lysed for western blot analysis at 48 h posttransfection and size-separated on a 10% reducing and denaturing SDS-PAGE gel. HA0 and HA1 were detected using a rabbit polyclonal H5N1 serum. Beta-actin and a protein ladder were included as for panel B. Representative blots of three independent experiments are depicted. (D) Lovo cells were cotransfected with plasmid coding for human, chicken, or duck furin and HA containing intermediate cleavage site motifs REKR, RKTR, and RKKR, and processed as described for panel C. (E) Quantification of HA cleavage percentages. Bars represent the means inferred from the quantification of western blots from three independent experiments. Error bars represent the standard deviations (SD).

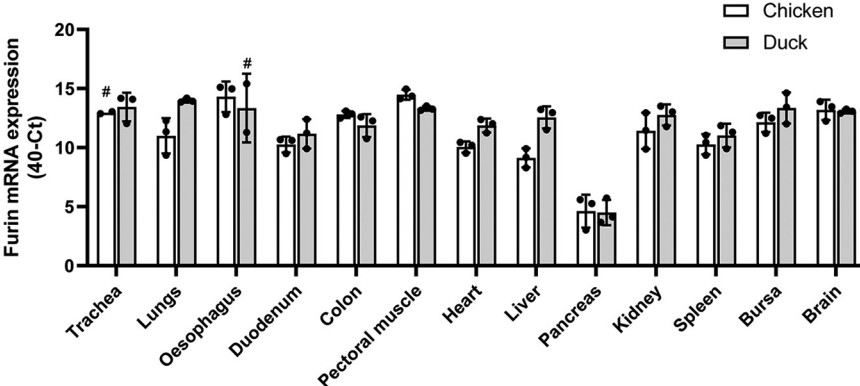

**FIG 3** Comparable levels of furin mRNA expression in tissues of chickens and ducks. Furin mRNA levels were determined by RT-qPCR on 10-ng input RNA of tissues of three donor animals. The hashtags indicate tissues for which only two donors were available. Individual data are shown as dots, bars represent the means, and the error bars show the SD.

Trypsin treatment also resulted in cleavage of HPAIV HA, but not as completely as during furin coexpression (Fig. 2C and E).

The consensus recognition motif of human furin is R-X-R/K-R, but its proteolytic cleavage efficiency increases in the presence of additional basic amino acids (51, 52). Therefore, the wild-type A/Indonesia/5/05 H5N1 MBCS motif might be a highly potent furin substrate due to the presence of a total of six basic residues. To assess whether a less-optimal furin cleavage site would also be cleaved by duck furin, we repeated the experiment using intermediate MBCS motifs REKR, RKTR, and tetrabasic cleavage site RKKR. The HA0 band remained the primary band detected upon transcomplementation of HA-REKR and HA-RKTR with all three species of furin, indicating inefficient cleavage (Fig. 2D and E). In contrast, furin from all three species efficiently cleaved the tetrabasic cleavage site RKKR, indicating that MBCS motifs containing only a few basic amino acids can also be cleaved by duck furin (Fig. 2D and E). Again, the reduction of total HA levels following cotransfection with avian furin was observed, as were low levels of background cleavage under the conditions for cotransfection with empty vector.

**Comparable ubiquitous expression of furin mRNA in adult chicken and duck tissues.** As HA-cleaving functionality was similar between chicken and duck furin in an overexpression system, we sought out to determine whether differences in furin expression patterns between the species might contribute to the observed differences in HPAIV tropism. The furin mRNA expression levels were determined in a range of different tissues from 6-week-old chickens (White Leghorn) and domestic ducks by reverse transcription-quantitative PCR (RT-qPCR). Furin mRNA was expressed ubiquitously in all tested duck tissues and to comparable extents as in chickens (Fig. 3). Efforts to support the current RNA-based expression analysis by protein quantification through immunohistochemistry failed due to the low endogenous furin levels and the need to use mammalian anti-furin antibody because of the lack of availability of specific avian anti-furin antibodies.

**Expression of PCs in duck endothelial cells coincides with multicycle replication of HPAIVs but not LPAIVs.** The systemic dissemination of HPAIVs in gallinaceous poultry, such as chickens, is determined by the ubiquitous expression of PCs. Systemic dissemination is greatly enhanced by the replication in vascular endothelial cells, the hallmark of HPAIV infection in chickens, which is generally absent in HPAIV-infected ducks (14). In the above-mentioned analysis, furin mRNA was expressed ubiquitously in all tested tissues from naive chicken and ducks (Fig. 3), but this analysis did not allow for the discrimination of PC mRNA expression between separate cellular subsets within tissue homogenates. Therefore, we continued the investigation of PC mRNA expression in ducks in the cell type that supports the systemic dissemination of HPAIVs in chickens, the endothelium.

Previously, we characterized viral replication kinetics and innate immune responses of primary duck aortic endothelial cells (dAEC) from embryonic origin upon HPAIV H5N1

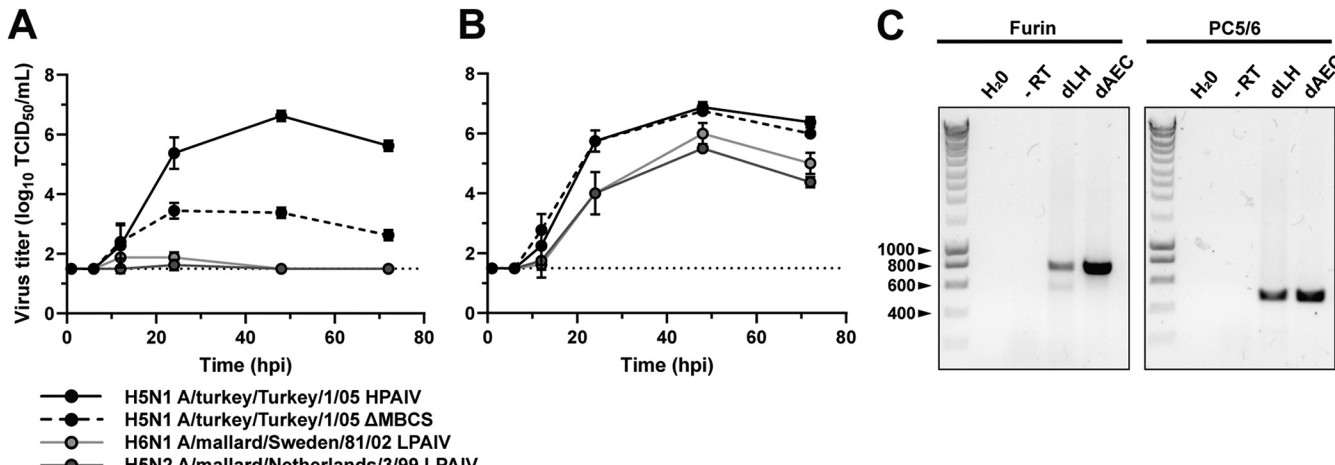

**FIG 4** Multicycle replication of HPAIVs but not LPAIVs in duck endothelial cells. (A and B) Primary duck aortic endothelial cells (dAEC) were inoculated with HPAIV H5N1 A/turkey/Turkey/1/05 or the ΔMBCS variant of A/turkey/Turkey/1/05, LPAIV H6N1 A/mallard/Sweden/81/02, or LPAIV H5N2 A/mallard/Netherlands/3/99 at an MOI of 0.001 in the absence (A) or presence (B) of TPCK-trypsin. Supernatants were harvested at the indicated time points, and infectious virus titers were determined by endpoint titration in MDCK cells and expressed as the $TCID_{50}$ per milliliter. Data are presented as means ± SD from two independent experiments. The dotted line represents the limit of detection of the endpoint titration assay. (C) RT-PCR for furin and PC5/6 expression in dAEC and embryonic duck lung homogenate (dLH) as positive control. The left lane is the DNA size marker and a selection of the base pair sizes is indicated on the left. $H_2O$, no-template negative control, for which water was added instead of cDNA; −RT, samples where RNA was used as the template as a negative control for genomic DNA detection.

A/Vietnam/1203/04 inoculation (25). Duck endothelial cells were permissive to HPAIV infection and could sustain multicycle replication, indicative of the expression of furin-like proteases. Here, we showed that another HPAIV strain, H5N1 A/turkey/Turkey1/05, replicated to high titers in duck endothelial cells in the absence of trypsin (Fig. 4A). In contrast, multiple LPAIVs (H5N1 ΔMBCS, H5N2, and H6N1) required the addition of exogenous trypsin for multicycle replication (Fig. 4B), indicating that primary dAEC do not express LPAIV-activating trypsin-like proteases. The mRNA expression of HPAIV-activating furin and the transmembrane and secreted isoforms of PC5/6 in duck endothelial cells was confirmed by RT-PCR (Fig. 4C).

## DISCUSSION

Cleavage of HA by host proteases is an essential step during the AIV replication cycle, influencing virus tropism, virulence, and pathogenesis. In chickens, presence of an MBCS in HA allows for AIV replication in a plethora of different tissues and cell types, including endothelial cells, resulting in systemic viral dissemination and severe hemorrhagic disease (12). In contrast, HPAIV infections in wild birds are generally milder, with viral replication confined to epithelial and parenchymal cells in specific tissues, excluding the endothelium (13–15, 36). HPAIV-activating ubiquitously expressed furin-like proteases are the host factors that allow for the systemic dissemination of HPAIVs in poultry, and therefore we hypothesized that differences in furin-like protease activity and/or expression contribute to the stark differences of HPAIV tropism between poultry and wild aquatic birds. Here, we showed that duck furin has a high sequence identity to chicken furin, is functionally capable of cleaving HPAIV HA, and that its mRNA is expressed in all tested duck tissues and in primary endothelial cells. Therefore, it is unlikely that differences in HA-activating protease expression largely contribute to the differences in HPAIV pathogenesis and tropism between chickens and ducks.

Furin mRNA was expressed in all 13 tested tissues as well as in primary duck aortic endothelial cells. PC5/6 mRNA was also detected in dAEC, in line with high mRNA expression levels of PC5/6 in rat and mouse aorta (53, 54) and the presence of mRNA in embryonic chicken endothelial cells (45). Although our detection primers targeted both the A and B isoforms of PC5, the soluble PC5A isoform is more likely to be expressed in duck endothelial cells than the membrane-anchored PC5B, as it is

ubiquitously expressed in murine tissues and was detected in the murine aorta, whereas PC5B was not (54). For final conclusions on ubiquitous furin expression, an analysis of protein expression levels is warranted, but our attempts in detecting endogenous furin levels in tissues were hampered due to the low assay sensitivity and the limited species cross-reactivity of furin antibodies.

Duck furin had moderate amino acid identity with human furin (79%), whereas it was highly similar to chicken furin (96%). The existing amino acid differences between chicken and duck furin did not include regions with appointed functions, except for the basolateral sorting signal that was EECE and EEYE. The functional relevance of this difference is unclear, but as HPAIV HA is activated in the TGN (55), prior to basolateral sorting, it is not likely that this difference would impact HPAIV activation. Of note are the amino acid differences between avian and human furin, as some recent publications have shown that multibasic HA cleavage sites were more efficiently cleaved in avian than in human cells (46, 56). Tse et al. demonstrated that mutant H9 HAs, with an RSKR cleavage site and removed glycosylation site on position 13, were more efficiently cleaved by chicken furin than by human furin and that chicken cell lines contained slightly higher furin mRNA levels than human and quail cell lines (46). Parvin et al. showed a 2- to 4-fold-higher cleavability of H5 HPAIV HA in chicken and quail cell lines than in human and swine cell lines. Detailed studies on the molecular determinants of these cleavage efficiency differences are warranted, perhaps taking into account the differences in the inhibitory prodomain and cytoplasmic tail. For example, the histidine content in the prodomain is correlated with pH-dependent PC activation: the presence of fewer histidine residues is associated with a lower activation pH, which is present later in the constitutive secretory pathway (57). Interestingly, the avian prodomain contains one more histidine than human furin, six versus five histidines, respectively, which might result in earlier activation within the TGN.

As expected based on the sequence identity, we showed here that duck furin cleaves wild-type HPAIV HA and tetrabasic HA-RKKR in an overexpression system. Cleavage efficiency of tetrabasic cleavage sites by furin varies per strain and is partially dependent on a glycosylation site at position 22 (H3 numbering), proximal to the cleavage loop, whose presence can reduce furin cleavage efficiency (58). Interestingly, the HA from H5 A/Indonesia/5/05 does contain that putative glycosylation site, but natural HPAIVs with a tetrabasic cleavage site and the glycosylation site have been described before (4). High furin levels resulting from the exogenous overexpression might allow for cleavage that would not occur with endogenous levels. Additionally, further research on the activity of endogenous furin-like proteases in the context of HA cleavage is warranted, as differences in furin activity might exist between species, tissues, and cell types, especially in the presence of endogenous furin inhibitors, such as guanylate-binding proteins 2 and 5, which can affect HA cleavage efficiency (59). Of note, we observed a reduction in total HA levels following furin transcomplementation. This phenomenon has been observed before in coexpression studies of HA of various subtypes with certain trypsin-like proteases and might be caused by protein overdigestion (60–62), but a definitive explanation is lacking.

The replication of HPAIVs in the endothelium is a hallmark of HPAIV infections in terrestrial poultry and most likely contributes to HPAIV systemic dissemination, pathogenesis, and high mortality (63–65). In ducks, endothelial cells are not positive for viral antigen, with the exception of ducks infected with H5N8 HPAIVs from 2016 (16, 22, 23), and endotheliotropism-associated clinical manifestations such as hemorrhage and edema are not observed (13, 15, 21, 36). Here, we concurrently observed the trypsin-independent replication of HPAIV, as was shown by us before (25), and expression of PC mRNA in duck endothelial cells. Additionally, we have shown that multicycle replication of LPAIVs in duck endothelial cells remained trypsin-dependent. LPAIV infection is not associated with endotheliotropism in avian species, which has been corroborated previously by the absence of trypsin-like proteases in chicken endothelial cells (66). Altogether, results from our study suggest that other intrinsic differences than protease expression and function between chickens and ducks,

**TABLE 1** List of primers and probes used in this study

| Assay and purpose | Target | Oligo | Sequence (5′–3′) |
|---|---|---|---|
| RT-PCR | | | |
| ORF sequencing | Furin section 1 | Forward | ATGGTGTGTCGGGCTGA |
| | | Reverse | CGTCACAATCTGCTTCTCGT |
| | Furin section 2 | Forward | CATCTACACGCTGTCCATCA |
| | | Reverse | CATAGTTGTTGGCGTCACTG |
| | Furin section 3 | Forward | AGCTGATGGCTTCAACGACT |
| | | Reverse | TCAAAGGGCACTTTGGTCTC |
| mRNA detection | Furin | Forward | CATCTACACGCTGTCCATCA |
| | | Reverse | CATAGTTGTTGGCGTCACTG |
| | PC5/6 isoform A/B | Forward | ATGGAACCAAAGGTGGAGTG |
| | | Reverse | ATCCAGCATCCGTACACCTC |
| RT-qPCR | | | |
| mRNA quantification | Furin | Forward | GAGGGRTACTACCTGCACCA |
| | | Reverse | AYGCTGTTCTCCAGGTTGTAGT |
| | | Probe[a] | 6-FAM–AGAGCTGCCTGAAGCGCTGCC–BHQ1 |

[a]6-FAM, 6-carboxyfluorescein; BHQ1, black hole quencher 1.

such as innate immune responses (24–28), are more likely to underlie the observed differences in HPAIV pathogenesis and tropism than furin-like protease expression and activity.

**Conclusion.** Here, we have sequenced and cloned furin from domestic ducks, which was identical to the amino acid sequence of furin from mallards, and we confirmed its activity as an HPAIV HA-cleaving protease. Furin mRNA was expressed ubiquitously, and the expression of the proprotein convertases furin and PC5/6 mRNA in duck endothelial cells coincided with HPAIV replication. It is therefore unlikely that differences in furin-like protease expression and activity largely contribute to the stark differences in HPAIV pathogenesis and tropism between chickens and ducks. Still, further research elucidating the details of endogenous furin protein expression and activity in various species, tissues, and cell types is warranted.

## MATERIALS AND METHODS

**Cell culture.** Primary duck aortic endothelial cells were isolated and characterized as previously described (25) and maintained in microvascular endothelial cell growth medium-2 (EGM-2MV; Lonza) on plates coated with 0.2% gelatin (Sigma-Aldrich) at 40℃ in 5% $CO_2$. Furin-deficient LoVo cells (CCL-229; ATCC) were maintained in Kaighn's modification of Ham's F-12 medium (Thermo Fisher Scientific) supplemented with 10% fetal calf serum (FCS; Sigma), 100 U/mL penicillin (Lonza), and 100 $\mu$g/mL streptomycin (Lonza), at 37℃ in 5% $CO_2$. Madin-Darby canine kidney cells (MDCK) were cultured in Eagle's minimal essential medium (Lonza) supplemented with 10% FCS, 100 U/mL penicillin, 100 U/mL streptomycin, 2 mM L-glutamine (Lonza), 1.5 mg/mL sodium bicarbonate (Lonza), 20 mM HEPES (Lonza), and nonessential amino acids (Lonza) at 37℃ in 5% $CO_2$. Duck lung homogenate (dLH) was obtained by digesting freshly isolated lungs of 21-day-old domestic duck embryonated eggs with collagenase and dispase (Roche) followed by treatment with a red blood cell lysis buffer (Roche) for 10 min at room temperature.

**Sequencing of the duck furin open reading frame.** The furin ORF was sequenced from mRNA of domestic ducks (*Anas platyrhynchos domesticus*) and mallards (*Anas platyrhynchos*). Total RNA was isolated from pooled embryonic mixed male and female lung cells or bone marrow by using the High Pure RNA isolation kit (Roche) according to the manufacturer's instructions. Concentration and quality of the RNA were determined using the NanoDrop spectrophotometer (Thermo Fisher Scientific). For cDNA synthesis, 500 ng RNA was reverse transcribed using random hexamer primers with SuperScript IV reverse transcriptase (Thermo Fisher Scientific) according to the manufacturer's instructions. The furin ORF was amplified in 3 sections using the PfuUltra II Fusion HS DNA polymerase (Agilent) and the primers listed in Table 1. PCR fragments were size-selected by gel electrophoresis, purified from gel using the MinElute gel extraction kit (Qiagen), and sequenced by Sanger sequencing (3500xl Genetic Analyzer, Applied Biosystems) using the same primers as during amplification.

**Generation of furin and HA expression plasmids.** Duck furin was cloned from embryonic lung homogenate into the multiple cloning site of the pCAGGs plasmid, a kind gift of A. Garcia-Sastre (Icahn School of Medicine, New York, USA). Human furin (Sino Biological catalog number HG10141-M; accession number NM_002569.2) and chicken furin (Genscript catalog number OGa00332; accession number NM_204715.1) were ordered and cloned into the pCAGGs vector. The HA gene segment of A/Indonesia/5/05, cloned in a modified version of the pHW2000 vector (67), was used as the template to introduce mutations into the cleavage site region by site-directed mutagenesis. The full HA segments were then cloned into pCAGGs.

**HA cleavage assessment by western blotting.** Furin-deficient LoVo cells were seeded at 70% confluence in 6-well dishes and transfected at 48 h using Lipofectamine 3000 (Thermo Fisher Scientific) transfection reagent, following the manufacturer's instructions. A 2.5-$\mu$g amount of expression plasmid encoding the wild-type or cleavage site mutants of the HA of A/Indonesia/5/05 was transfected, together with 2.5 $\mu$g plasmid encoding human, chicken, or duck furin or empty vector pCAGGs. At 48 h, cells were washed with phosphate-buffered saline (PBS) and scraped from the surface before collection in an Eppendorf tube. Indicated samples were treated with 2.5 $\mu$g/mL TPCK-treated trypsin (Sigma-Aldrich) in PBS for 10 min at 37°C. Following centrifugation, cell pellets were lysed in hot lysis buffer (1% sodium dodecyl sulfate [SDS], 100 mM NaCl, 10 mM EDTA, 10 mM Tris-Cl; pH 7.5) with cOmplete Mini protease inhibitor cocktail (Merck), and DNA was sheared by repeated pipetting through a 29-gauge needle. Total protein concentration was determined with Pierce bicinchoninic acid (Thermo Fisher Scientific) quantification. When indicated, samples were deglycosylated, to allow for the optimal comparison of protein size and content, using peptide-$N$-glycosidase F (New England Biolabs) according to the manufacturer's instructions. Samples were denatured and reduced for 10 min at 95°C in loading buffer (20% glycerol, 4% SDS, 125 mM Tris-HCl, 3 M bromophenol blue) supplemented with 10% $\beta$-mercaptoethanol. Per sample, a total of 6 $\mu$g of protein was run on a denaturing and reducing SDS-PAGE gel (4.2% stacking gel, 10% running gel) and transferred to a 0.45-$\mu$m nitrocellulose membrane (Amersham Protran) for 1 h in transfer buffer (191.8 mM glycine, 25 mM Tris base, 20% methanol). The Precision Plus Protein Kaleidoscope marker (Bio-Rad) was run alongside the samples. Membranes were blocked overnight at 4°C in PBS with 5% (wt/vol) dried nonfat milk and 0.1% Tween and incubated with a 1:1,000 dilution of serum raised in rabbits against A/Hong Kong/156/97 H5N1 or with 2 $\mu$g/mL anti-furin antibody (ab183495; Abcam) in blocking buffer for 2 h at room temperature. Membranes were washed thrice with PBS–0.1% Tween for 5 min and incubated with 1:2,000 swine-anti-rabbit–horseradish peroxidase (HRP; Agilent) and 1:20,000 anti-beta-actin–HRP (Bio-connect) in blocking buffer for 1 h at room temperature. The blot was developed using ECL Plus detection reagents (Merck) following the manufacturer's instructions and imaged on a ChemiDoc MP (Bio-Rad). Protein size was inferred from the Precision Plus Protein Kaleidoscope marker on the direct light picture of the blot. Relative HA cleavage efficiencies were calculated based on the pixel intensities of each band, as measured with Fiji software, using the following formula: $[HA_1/(HA_0 + HA_1)] \times 100\%$.

**Viral replication kinetics.** HPAIV A/turkey/Turkey/1/05 (H5N1), A/turkey/Turkey/1/05 ΔMBCS with cleavage site reverted to LPAIV consensus sequence RETR, LPAIV A/mallard/Netherlands/3/99 (H5N2), and LPAIV A/mallard/Sweden/81/02 (H6N1), in their respective genetic backgrounds, were rescued by reverse genetics as previously described (67). The viruses were propagated in MDCK cells to produce virus stocks (2 passages) and sequenced using Sanger sequencing to confirm authenticity. LPAIV A/mallard/Sweden/81/02 (H6N1) was additionally propagated in 11-day-old embryonated chicken eggs. Infectious virus titers of the virus stocks were determined by endpoint titration in MDCK cells in the presence of 0.4 $\mu$g/mL TPCK-treated trypsin, as previously described (68), and expressed as the median tissue culture infectious dose ($TCID_{50}$) per milliliter, as calculated from 8 replicates according to the method of Spearman-Karber (69). For replication kinetics assessment, confluent monolayers of dAEC were inoculated at a multiplicity of infection (MOI) of 0.001. After 1 h of incubation, the inoculum was removed, and the cells were washed thrice with PBS. Fresh serum-free EGM-2MV with or without 0.14 $\mu$g/mL TPCK-treated trypsin was overlaid, and cultures were incubated at 40°C in 5% $CO_2$. Supernatants were harvested at the specified time points and stored at $-80$°C until further analysis. Infectious virus titers in the supernatant were determined by endpoint titration in MDCK cells in the presence of 0.4 $\mu$g/mL TPCK-treated trypsin and calculated from 4 replicates. All infection experiments were performed in biosafety level 3 containment facilities at the Erasmus Medical Center, Rotterdam, the Netherlands.

**Assessment of furin mRNA expression in tissues from chickens and ducks.** Tissues were obtained from 5- to 6-week-old female White Leghorn chickens and male and female domestic ducks. Tissues were collected in transport medium containing Eagle's MEM with Hanks' balanced salt solution, 25 mM HEPES, 10% glycerol, 0.5% lactalbumin hydrolysate (Sigma-Aldrich), 100 U/mL polymyxin B sulfate (Sigma-Aldrich), 100 U/mL nystatin (Sigma-Aldrich), 50 mg/mL gentamicin (Gibco), 100 U/mL penicillin, and 100 $\mu$g/mL streptomycin, homogenized using ceramic lysing spheres (MP Biomedicals), and stored at $-80$°C. Total nucleic acid isolations were performed using an in-house developed method as previously described (70). Briefly, 100 $\mu$L of sample was added to 150 $\mu$L MagNa Pure 96 external lysis buffer (Roche). This mixture was added to 50 $\mu$L AMPure XP magnetic beads (Beckman Coulter) and incubated for 15 min at room temperature. The plate was placed on a magnetic block (DynaMag 96 side skirted magnet; Thermo Fisher Scientific) to allow for the beads with bound nucleic acids to displace toward the magnet. The beads were washed thrice with 70% ethanol, air dried for 1 min, and eluted in 50 $\mu$L PCR-grade water. To eliminate genomic DNA contamination, the eluate was treated with Turbo DNase (Ambion) following the manufacturer's instructions. The RNA concentration was determined using the Qubit RNA high sensitivity kit (Thermo Fisher Scientific). The RNA concentration was set to 2 ng/$\mu$L, and 10 ng was used as input for RT-qPCR using primers for furin (Table 1) and 4× TaqMan Fast Virus 1-Step master mix (Thermo Fisher Scientific). Amplification and detection were performed on an ABI7500 system (Thermo Fisher Scientific) using the following cycling program: 5 min at 50°C, 20 s at 95°C, and then (3 s at 95°C, 31 s at 60°C) for 45 cycles. PCR efficiency, linear range, and sensitivity were determined for the novel furin primer set. Furin mRNA levels were determined for 3 animals per species, or as specified otherwise, and depicted as 40 $-$ the cycle threshold ($C_T$).

**Detection of PC mRNA by RT-PCR.** Total RNA was isolated from dAEC and dLH using the High Pure RNA isolation kit (Roche) according to the manufacturer's instructions. For cDNA synthesis, 100 ng of RNA was reverse-transcribed using random primers (Promega) and SuperScript IV reverse transcriptase. Furin and PC5/6 mRNA expression was determined using primers listed in Table 1 and the AmpliTaq

Gold DNA polymerase kit (Thermo Fisher Scientific) according to the manufacturer's instructions. The following cycling program was used: 1 min at 95°C, (20 s at 95°C, 30 s at 57°C, 1 min at 72°C) for 30 cycles, and 3 min 72°C. As negative controls, 100 ng RNA, to exclude the detection of genomic DNA, or water was used as input. PCR product sizes were analyzed by gel electrophoresis, and the amplicons were sequenced using Sanger sequencing to confirm PCR specificity.

**Ethics statement.** Experiments involving the harvest of tissues from adult animals were conducted according to the European guidelines (EU directive on animal testing 86/609/EEC) and Dutch legislation (Experiments on Animals Act, 1997). The experimental protocols 18-6744-02, 18-6744-04, 18-6744-05, 18-6744-07 under license number AVD1010020186744 were reviewed and approved by an ethical review committee at the Erasmus Medical Center, Rotterdam, the Netherlands. Ethical review was waived for the experiments involving embryos, because embryos (regardless of the gestational stage) from avian species are not subjected to ethical regulations in the European Union. (See the "European directive on the protection of animals used for scientific purposes," Article 1, point 3, in https://eur-lex.europa.eu/legal-content/EN/TXT/HTML/?uri=CELEX:32010L0063&from=EN; accessed 12 July 2022.)

## ACKNOWLEDGMENTS

This work has received funding from the European Union's Horizon 2020 research and innovation program under DELTA-FLU, grant agreement 727922, and NIAID NIH contract numbers HHSN272201400008C and 75N93021C00014.

We thank Alexander Gultyaev for his insights on genome annotation.

We declare no competing interests.

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
