## [Reviewer comments · Microbiology Spectrum]

Microbiology Spectrum

Conserved Expression and Functionality of Furin between Chickens and Ducks as Activating Protease of Highly Pathogenic Avian Influenza Virus Hemagglutinins

Anja de Bruin, Monique Spronken, Theo Bestebroer, Ron Fouchier, and Mathilde Richard

Corresponding Author(s): Mathilde Richard, Erasmus MC

Review Timeline:

Submission Date:	November 10, 2022
Editorial Decision:	December 22, 2022
Revision Received:	February 9, 2023
Accepted:	February 23, 2023

Editor: Daniela Rajao

Reviewer(s): The reviewers have opted to remain anonymous.

Transaction Report:

DOI: <https://doi.org/10.1128/spectrum.04602-22>

December 22, 2022

Dr. Mathilde Richard
Erasmus MC
Department of Viroscience
Dr Molewaterplein 50
Rotterdam 3015 GE
Netherlands

Re: Spectrum04602-22 (Conserved Expression and Functionality of Furin between Chickens and Ducks as Activating Protease of Highly Pathogenic Avian Influenza Virus Hemagglutinins)

Dear Dr. Mathilde Richard:

Thank you for submitting your manuscript to Microbiology Spectrum. Your manuscript has now been reviewed by experts in the field. Although all reviewers agree this is an important and timely topic, they pointed some crucial information is needed regarding the furin expression and activity in both species.

Link Not Available

Sincerely,

Daniela Rajao

Journals Department
Reviewer comments:

Reviewer #1 (Comments for the Author):

De Bruin and colleagues analyzed expression of duck furin and its ability to cleave hemagglutinin (HA) of highly pathogenic avian influenza viruses (HPAIV). They show that the amino acid sequence of duck furin is very similar to that of chicken furin and provide evidence that duck and chicken furin can cleave HA and have similar requirements regarding the cleavage site sequence. Further, they demonstrate ubiquitous and comparable expression of furin in duck and chicken tissues and show that proteolytic cleavage limits spread of LPAIV in duck endothelial cells, which are reported to express furin. The results are of

interest to the field. Some points remain to be addressed.

Major

The relative expression of transfected chicken and duck furin is important for the interpretation of the results and the respective immunoblot (and if possible IF) data should be shown.

Minor

"Cleavage enables the fusion peptide, located at the N-terminus of HA2, to undergo a conformational change upon exposure" It might be more correct to state that HA2 undergoes a conformational change.

Guanylate-Binding Proteins (GBP) 2 and 5 block cleavage of viral glycoproteins by furin. The authors might want to speculate whether different expression and/or activity of GBP proteins in waterfowl and poultry might contribute to the difference in HPAIV tropism and virulence in these animals.

Reviewer #2 (Comments for the Author):

The manuscript by Bruin et al. addresses the interesting question of whether the virus-activating protease furin differs in tissue distribution and its capability to cleave influenza A virus hemagglutinin (HA) in duck versus chicken and thereby contributes to the differences in pathogenicity of highly pathogenic avian influenza viruses (HPAIV) in these avian species.

The authors show that upon transient co-expression both duck and chicken furin cleave HA of HPAIV. Further, using qRT-PCR analysis the authors show that furin mRNA expression is similar in various tissues of chicken and duck. The authors conclude that furin function and expression is comparable between chicken and duck and therefore it is unlikely that furin largely contributes to the differences in HPAIV pathogenesis and tissue tropism in chicken versus duck.

The manuscript is well written, the data are solid. However, the authors did not further characterize the substrate specificity of duck versus chicken furin nor investigated whether furin is expressed as an active enzyme in different tissues of duck versus chicken. These investigations would still be important in order to be able to definitively assess whether or not HA cleavage by furin plays a role in HPAIV pathogenesis in duck versus chicken. The conclusion that furin does not contribute to differences in HPAIV pathogenesis in duck versus chicken cannot be supported by only showing that both furins are capable of cleaving HPAIV HA in vitro and that furin mRNA is present in various tissues.

The following points need to be considered:

Fig. 3) The figure shows levels of furin mRNA, but no expression or activity of the protein itself. This is a notable limitation of the study. Endogenous furin inhibitors could result in differences in furin activity in duck versus chicken and thereby in HPAIV activation in both species. The authors should show that furin activity is similar in chicken and duck tissues.

Moreover, using transient expression of chicken and duck furin the authors should examine whether both proteins have a comparable substrate specificity and/or sensitivity to protease inhibitors. A study by El Najjar et al. analyzed furin-like enzymes in fruit bat cells and indicated that subtle differences in activity of furin orthologs may exist between different mammalian species (<http://www.ncbi.nlm.nih.gov/pubmed/25706132>). This might also be true for different avian species.

Fig. 2B and 2C: Only low amounts of HA are detected upon co-expression of chicken furin or duck furin. Is there an explanation?

Staff Comments:

Preparing Revision Guidelines

- Point-by-point responses to the issues raised by the reviewers in a file named "Response to Reviewers," NOT IN YOUR COVER LETTER.

- Upload a compare copy of the manuscript (without figures) as a "Marked-Up Manuscript" file.
- Each figure must be uploaded as a separate file, and any multipanel figures must be assembled into one file.
- Manuscript: A .DOC version of the revised manuscript
- Figures: Editable, high-resolution, individual figure files are required at revision, TIFF or EPS files are preferred

Please return the manuscript within 60 days; if you cannot complete the modification within this time period, please contact me. If you do not wish to modify the manuscript and prefer to submit it to another journal, please notify me of your decision immediately so that the manuscript may be formally withdrawn from consideration by Microbiology Spectrum.

Review of the manuscript by Anja C.M. de Bruin *et al.*

The work is very topical given the unusually high numbers of HPAIV outbreaks reported in wild and domestic bird populations in Europe and elsewhere during the past few years. Research focusing on virus-avian host interactions are much appreciated since several gaps in our knowledge remain.

In this work, Anja C.M. de Bruin and colleagues addresses the hypothesis that differences in chicken and duck furin, and/or its expression, is the underlying reason for the observed differences in HPAIV pathogenicity in the two host species. Their approach applying sequencing, cloning, *in vitro* infections assays and studying mRNA expression levels is sound and makes their findings credible.

The article is clear, very well written and structured. The material and methods section is clear and sufficiently detailed. The figures are easy to follow and support the conclusions. I can warmly recommend this manuscript for publication with very minor comments to address.

Review

Row 22: Abstract. Highly pathogenic avian influenza viruses (HPAIVs) *generally/typically* emerge.... (or similar rephrasing to indicate that the spillover event is not an obligatory step for conversion)

Row 25: ...the proteolytic cleavage site in the hemagglutinin (HA, *the viral fusion protein*)...

Row 58: Please explain HA, e.g viral glycoprotein responsible for fusion/entry

Row 127: I'm happy to read that a comparison with the equivalent mallard sequence was made. Were there differences on the nucleotide level and why did you choose to work on the domesticated (and inbred) duck instead of the mallard in first place?

Response to Reviewers

We thank the reviewers for their constructive critiques. Please find their comments reproduced below, followed up by each of our answers in a green font.

Reviewer #1

De Bruin and colleagues analyzed expression of duck furin and its ability to cleave hemagglutinin (HA) of highly pathogenic avian influenza viruses (HPAIV). They show that the amino acid sequence of duck furin is very similar to that of chicken furin and provide evidence that duck and chicken furin can cleave HA and have similar requirements regarding the cleavage site sequence. Further, they demonstrate ubiquitous and comparable expression of furin in duck and chicken tissues and show that proteolytic cleavage limits spread of LPAIV in duck endothelial cells, which are reported to express furin. The results are of interest to the field. Some points remain to be addressed.

Major

The relative expression of transfected chicken and duck furin is important for the interpretation of the results and the respective immunoblot (and if possible IF) data show be shown.

We thank the reviewer for their suggestion. We have now included the data from the furin immunoblot experiment, which was referred to as 'data not shown' in lines 164-165 of the original manuscript, in which we show that the expression of human, chicken, and duck furin upon transfection in LoVo cells is indeed similar (included in Figure 2, panel B)(described in lines 152-155). A comparison based on immunofluorescence was also attempted, but the species cross-reactivity of the furin antibody was suboptimal for analyzing 3D epitopes and therefore had to be discontinued. Similar issues were encountered during the attempts to quantify furin protein levels in avian tissues, which is explained in lines 189-192.

Minor

"Cleavage enables the fusion peptide, located at the N-terminus of HA2, to undergo a conformational change upon exposure" It might be more correct to state that HA2 undergoes a conformational change.

The sentence has been adjusted accordingly (line 77-79): "Cleavage enables the low-pH induced conformational change of HA and liberates the fusion peptide, located at the N-terminus of HA2, so that it is accessible to mediate fusion of the viral and endosomal membranes necessary for release of the genome into the cytoplasm".

Guanylate-Binding Proteins (GBP) 2 and 5 block cleavage of viral glycoproteins by furin. The authors might want to speculate whether different expression and/or activity of GBP proteins in waterfowl and poultry might contribute to the difference in HPAIV tropism and virulence in these animals.

We thank the reviewer for this suggestion. We completely agree with the potential effect that furin inhibitors might have on virus replication in different species, tissues, and cell types. To accommodate these thoughts, we have added the following to the discussion “Additionally, further research on the activity of endogenous furin-like proteases in the context of HA cleavage is warranted as differences in the percentage of active furin might exist between species, tissues, and cell types especially in the presence of endogenous furin inhibitors, such as Guanylate-Binding Proteins 2 and 5, which can affect HA cleavage efficiency” (lines 262-265) with referencing the main GBP2/5 paper describing the inhibitory effect on influenza replication (PMID: 31091448).

Reviewer #2

The manuscript by Bruin et al. addresses the interesting question of whether the virus-activating protease furin differs in tissue distribution and its capability to cleave influenza A virus hemagglutinin (HA) in duck versus chicken and thereby contributes to the differences in pathogenicity of highly pathogenic avian influenza viruses (HPAIV) in these avian species.

The authors show that upon transient co-expression both duck and chicken furin cleave HA of HPAIV. Further, using qRT-PCR analysis the authors show that furin mRNA expression is similar in various tissues of chicken and duck. The authors conclude that furin function and expression is comparable between chicken and duck and therefore it is unlikely that furin largely contributes to the differences in HPAIV pathogenesis and tissue tropism in chicken versus duck.

The manuscript is well written, the data are solid. However, the authors did not further characterize the substrate specificity of duck versus chicken furin nor investigated whether furin is expressed as an active enzyme in different tissues of duck versus chicken. These investigations would still be important in order to be able to definitively assess whether or not HA cleavage by furin plays a role in HPAIV pathogenesis in duck versus chicken. The conclusion that furin does not contribute to differences in HPAIV pathogenesis in duck versus chicken cannot be supported by only showing that both furins are capable of cleaving HPAIV HA in vitro and that furin mRNA is present in various tissues.

The following points need to be considered:

Fig. 3) The figure shows levels of furin mRNA, but no expression or activity of the protein itself. This is a notable limitation of the study. Endogenous furin inhibitors could result in differences in furin activity in duck versus chicken and thereby in HPAIV activation in both species. The authors should show that furin activity is similar in chicken and duck tissues.

We agree with reviewer #2 that differences in furin expression and activity in the context of HA cleavage in different cell types from different tissues might play a role. We did attempt to quantify protein expression levels in avian tissues, but were hampered by low endogenous furin levels and the lack of availability of anti-avian furin antibodies, as described in lines 189-192, and we have additionally discussed this issue in lines 234-236. Furin enzyme activity is usually measured using small peptides, including in the paper the reviewer references in the comment below (El Najjar et al.). However, this does unfortunately not directly correlate with furin cleavage activity in the context of HA due to the influence of 3D structure and steric hindrance of surrounding structures as shown

by Parvin et al. (PMID: 32234073). Furthermore, the localization of active furin within cellular organelles is important in the context of HA activation and this co-localization of enzyme and substrate is lost in methods where crude cell lysates are analyzed for furin activity. Therefore, the experiments suggested by the reviewer are interesting but still do not yield the necessary data to draw stronger conclusions than those in the current version of the manuscript. Yet, we agree that endogenous furin activity and the role of furin-inhibitors (also in light of the comment from Reviewer #1 regarding GBP2/5) must be emphasized stronger in the discussion and therefore a section on this topic has been added (lines 262-265) to accommodate the reviewer. Furthermore, we have nuanced our conclusions in the manuscript (lines 39 and 290-292).

Moreover, using transient expression of chicken and duck furin the authors should examine whether both proteins have a comparable substrate specificity and/or sensitivity to protease inhibitors. A study by El Najjar et al. analyzed furin-like enzymes in fruit bat cells and indicated that subtle differences in activity of furin orthologs may exist between different mammalian species (<http://www.ncbi.nlm.nih.gov/pubmed/25706132>). This might also be true for different avian species.

The substrate specificity of human, chicken, and duck furins was already investigated here in the context of HA, which is the relevant substrate for these proteases in relation to influenza virus replication. The data are shown in Figure 2 where we compared the specificity of the human, chicken, and duck furins for different substrates, the monobasic cleavage site RETR, intermediate cleavage sites RKTR and REKR and the tetrabasic cleavage site RKKR. The cleavage percentage of both the intermediate cleavage sites did not differ significantly from RETR, whereas HA-RKKR cleavage was almost complete and resembled that of a wild-type multibasic cleavage site. Testing the effect of protease inhibitors on furin from different species is interesting, but outside the scope of this manuscript.

Fig. 2B and 2C: Only low amounts of HA are detected upon co-expression of chicken furin or duck furin. Is there an explanation?

The phenomenon of HA protein levels decreasing during protease co-transfection has been described in several articles before, as mentioned in the discussion (line 265-269), but a definitive explanation is lacking. The reduction in total HA protein has been suggested to be caused by overdigestion following protease overexpression. However, this has, to our knowledge, not been properly studied. To accommodate the reviewer, the overdigestion hypothesis has been added (line 267-268) and an additional two articles that describe this phenomenon have been cited (PMID:19158246 and PMID:19840668).

Reviewer #3

The work is very topical given the unusually high numbers of HPAIV outbreaks reported in wild and domestic bird populations in Europe and elsewhere during the past few years. Research focusing on virus-avian host interactions are much appreciated since several gaps in our knowledge remain.

In this work, Anja C.M. de Bruin and colleagues addresses the hypothesis that differences in chicken and duck furin, and/or its expression, is the underlying reason for the observed differences in

HPAIV pathogenicity in the two host species. Their approach applying sequencing, cloning, *in vitro* infections assays and studying mRNA expression levels is sound and makes their findings credible.

The article is clear, very well written and structured. The material and methods section is clear and sufficiently detailed. The figures are easy to follow and support the conclusions. I can warmly recommend this manuscript for publication with very minor comments to address.

We thank the reviewer for their kind words regarding the experimental methods, obtained data, and conclusions from our article.

Review

Row 22: Abstract. Highly pathogenic avian influenza viruses (HPAIVs) *generally/typically* emerge... (or similar rephrasing to indicate that the spillover event is not an obligatory step for conversion)

The sentence has been adjusted accordingly (line 22).

Row 25: ...the proteolytic cleavage site in the hemagglutinin (HA, *the viral fusion protein*)...

An explanation of HA has been added "...in the viral binding and fusion protein, the hemagglutinin (HA), ..." (line 25-26).

Row 58: Please explain HA, e.g viral glycoprotein responsible for fusion/entry

Explanation for both HA and NA have been added (line 59-61): "Influenza A viruses are classified based on the antigenic properties of the two surface glycoproteins, the hemagglutinin (HA), mediating receptor binding and membrane fusion, and the neuraminidase (NA), mediating virion release".

Row 127: I'm happy to read that a comparison with the equivalent mallard sequence was made. Were there differences on the nucleotide level and why did you choose to work on the domesticated (and inbred) duck instead of the mallard in first place?

We chose to work with the domesticated duck species rather than the mallard duck species due to the difference in year-round accessibility. Our analyses required access to 6-week-old domestic ducks for the purpose of naïve tissue harvest and to embryonated duck eggs for the purpose of primary endothelial cell culture. Mallard ducks were not easily available, whereas domestic ducks were, due to their use for meat production. Furthermore, it has been shown that the infection kinetics upon HPAIV inoculations are comparable between domestic and mallard ducks (PMID:16103179). Therefore, using domestic ducks as a proxy for mallard ducks is justified. Although some nucleotide differences between domestic and mallard duck furin were observed, it was not possible to stratify these due to a limited number of analyzed individual birds as well as the pooling of individuals, especially in the context of single nucleotide polymorphisms.

February 23, 2023

Dr. Mathilde Richard
Erasmus MC
Department of Viroscience
Dr Molewaterplein 50
Rotterdam 3015 GE
Netherlands

Re: Spectrum04602-22R1 (Conserved Expression and Functionality of Furin between Chickens and Ducks as Activating Protease of Highly Pathogenic Avian Influenza Virus Hemagglutinins)

Dear Dr. Mathilde Richard:

Your manuscript has been accepted, and I am forwarding it to the ASM Journals Department for publication. You will be notified when your proofs are ready to be viewed.

Sincerely,

Daniela Rajao
Editor, Microbiology Spectrum
